# Diagnosis and Management of Glioblastoma: A Comprehensive Perspective

**DOI:** 10.3390/jpm11040258

**Published:** 2021-04-01

**Authors:** Vianney Gilard, Abdellah Tebani, Ivana Dabaj, Annie Laquerrière, Maxime Fontanilles, Stéphane Derrey, Stéphane Marret, Soumeya Bekri

**Affiliations:** 1Department of Neurosurgery, Rouen University Hospital, 76000 Rouen, France; stephane.Derrey@chu-rouen.fr; 2Normandie Univ, UNIROUEN, CHU Rouen, INSERM U1245, 76000 Rouen, France; abdellah.tebani@chu-rouen.fr (A.T.); annie.laquerriere@chu-rouen.fr (A.L.); soumeya.bekri@chu-rouen.fr (S.B.); 3Department of Metabolic Biochemistry, Rouen University Hospital, 76000 Rouen, France; ivana.dabaj@chu-rouen.fr (I.D.); stephane.marret@chu-rouen.fr (S.M.); 4Department of Neonatology, Pediatric Intensive Care, and Pediatric Neurology, Rouen University Hospital, 76000 Rouen, France; 5Department of Pathology, Rouen University Hospital, 76000 Rouen, France; 6Department of Medical Oncology, Cancer Centre Henri Becquerel, 76038 Rouen, France; maxime.fontanilles@chb.unicancer.fr

**Keywords:** glioblastoma, molecular pathology, omics, pathogenesis, personalized therapies

## Abstract

Glioblastoma is the most common malignant brain tumor in adults. The current management relies on surgical resection and adjuvant radiotherapy and chemotherapy. Despite advances in our understanding of glioblastoma onset, we are still faced with an increased incidence, an altered quality of life and a poor prognosis, its relapse and a median overall survival of 15 months. For the past few years, the understanding of glioblastoma physiopathology has experienced an exponential acceleration and yielded significant insights and new treatments perspectives. In this review, through an original R-based literature analysis, we summarize the clinical presentation, current standards of care and outcomes in patients diagnosed with glioblastoma. We also present the recent advances and perspectives regarding pathophysiological bases as well as new therapeutic approaches such as cancer vaccination and personalized treatments.

## 1. Introduction

Glioblastoma is the most frequent primary malignant brain tumor in adults [1]. Over the last 20 years, a growing incidence of glioblastomas has been observed due to an increase in general population life expectancy and a better access to more accurate diagnostic tools such as MRI [2,3,4]. In a recent article based on a prospective registry of central nervous system (CNS) tumors in the US [5], glioblastoma accounts for 54% of brain gliomas with an annual incidence of 3.19 per 100,000. The peak of incidence occurs in the 6th decade for glioblastomas isocitrate dehydrogenase (IDH) wild type and earlier, around the 4th–5th decade, for glioblastomas IDH mutant [6]. Despite many therapeutic trials [7,8,9] and advances in the management of these tumors, the median survival remains poor, approximately 14 to 20 months, with a 5% five-year survival rate depending on age at diagnosis, molecular characteristics and management [10]. Due to the poor outcomes of this pathology, there is a need for a better understanding of the pathogenesis and metabolic pathways of glioblastomas. For a few years, advances in the different areas brought new insights and new therapeutic perspectives. Precision medicine, through the surge of omics technologies, offers new approaches for various oncology diseases [11,12]. Its main goal is to set a personalized perspective of the disease, taking into account individual variability of the patient along with his environment and predisposition. This approach requires exhaustive collection of glioblastomas data. This work proposed an exhaustive semi-automated review of literature regarding clinical presentation, diagnosis and management of patients presenting with glioblastoma. Moreover, we also emphasize the novelties and perspectives concerning this disease within the precision medicine era.

## 2. Materials and Methods

### 2.1. Literature Analysis

In this study, we performed a programmatic literature search for a more efficient and reproducible review process using the Adjutant R package [13]. We searched for articles related to glioblastoma, glioma physiopathology, diagnosis and treatments that were published between January 1990 and February 2020. We used only one query (glioblastoma). The resulting document corpus included articles metadata: PubMed IDs, year of publication, authors, article titles, article abstracts and any associated Medical Subject Heading (MeSH) terms. Titles and abstracts were decomposed into single terms, stemmed and filtered by Adjutant package. The term frequency inversed document frequency metrics for each term and created a sparse Document Term Matrix (DTM) for further analysis. t-Distributed Stochastic Neighbor Embedding (t-SNE) and hdbscan algorithms were used to perform unsupervised clustering using DTM data. The coordinates generated by t-SNE were used in the hdbscan algorithm to derive the topic clusters. Each cluster was then assigned a topic by using the five most frequent terms within the cluster.

### 2.2. Manual Curation: Inclusion and Exclusion Criteria

Following the topic clustering step, we validated our clusters using external manual curation assessing the correspondence between articles and cluster topics. Each sampled article was examined and either considered acceptable for further analysis or rejected. Inclusion criteria were topic relevance assessed by a neurosurgeon specialized in neuro-oncology; article in English; human data; original research or clinical trial. We further refined the corpus and cluster naming. Appendix A contains a list of all the articles along with their corresponding cluster.

### 2.3. Data Analysis

All the data analysis and visualization have been done using the R software [14].

## 3. Results

### 3.1. Literature Mining and Topic Clusters

The first-round analysis generated a document corpus of 2799 articles related to glioblastoma published over the past 30 years (Appendix A). Using article titles and abstracts, we derived topic clusters in an unsupervised manner, and classified articles according to their clustering status. Articles that never formed part of a cluster were removed from further analysis, leaving 1314 documents that formed 27 clusters. Cluster topics were assigned using the five most frequent terms within the cluster along with a manual curation of the included articles (Figure 1). The full list of articles and related clusters are presented in Appendix A.

### 3.2. Pathogenesis of Glioblastoma

#### 3.2.1. Risk Factors for High Grade Gliomas

Risk factors for glioblastoma onset are still unknown and studies dealing with this question frequently lack power. The exposure to ionizing radiation for the treatment of malignancy during infancy [15] is a very rare risk factor for the onset of glioma. The risk of developing a brain tumor after radiotherapy is increased if the radiation occurs at a younger age (<5 years) and seems volume- and dose-related but with no clear reported threshold [16,17]. The increased incidence of glioblastoma [18,19] raises the issue of environmental risk factors. The association between the use of mobile phone and the occurrence of glioblastoma has been reported in a meta-analysis [20]. However, these results are discordant and have been challenged in other studies [21,22]. The role of exposure to smoking or carcinogenic agents has been studied with no proven association to glioblastoma [23,24]. In very few cases (<1%), there is a genetic predisposition to glioblastoma development in patients with Lynch, Turcot type 1 or Li Fraumeni syndromes [25].

#### 3.2.2. Clinical Presentation

Clinical presentation depends on the tumor location and size at diagnosis. The most common presentation at diagnosis is a headache and/or nausea in a context of a large tumor or significant edema. Symptoms related to intracranial hypertension represent 30% of clinical signs followed by motor deficit (20%), loss of body weight and condition (17%), confusion (15%) and visual or speech deficit (13%) [26]. Epilepsy is not uncommon (15–20%) and easily controlled by anticonvulsant therapies. Epilepsy is associated with better outcomes probably due to the cortical location of glioblastomas presenting with seizures [27]. These symptoms are often associated and lead to a diagnosis in the weeks or months following their onset. An overview of clinical presentation at diagnosis is summarized in Figure 2. Glioblastomas are located in the supra-tentorial space in more than 85% of the cases explaining the above-mentioned symptoms and very infrequently in the brainstem and spinal cord (<5% each) or in the cerebellum (<3%). Up to 25% of glioblastomas occur in the frontal lobe [28], the largest cerebral lobe, and therefore are responsible for mood and executive disabilities in 15% of patients [26].

#### 3.2.3. Radiological Characteristics

Typical magnetic resonance imaging (MRI) characteristics of glioblastomas are well known [2,28,29,30,31]. They consist in infiltrative, heterogeneous intraparenchymal lesions which arise and spread from the white matter. Corpus callosum involvement is common. Glioblastomas are poorly circumscribed and display contrast enhancement at their margin as a sign of blood–brain barrier disruption. The center of the lesion is hypointense on T1-weighted images due to necrosis and the lesion is surrounded by brain edema which appears hyperintense on T2-weighted and fluid-attenuated inversion recovery (FLAIR) images. Diffusion-weighted images and apparent diffusion coefficients can provide valuable information concerning the suspected degree of malignancy of astrocytic tumors. More recent multimodal MRI techniques such as diffusion/perfusion sequences have provided supplementary information about the characteristics of the lesion itself and have enabled a more accurate diagnosis. Perfusion weighted imaging (PWI) reveals an increase in the cerebral blood flow corresponding to neoangiogenesis and blood–brain barrier disruption. MR spectroscopy in glioblastomas is characterized by an increase in choline/N-acetylaspartate and choline/creatinine ratios. Nevertheless, these characteristics reflect cellular activity and are not, when isolated, specific enough to diagnose glioblastoma. In addition, an elevated peak of lactate and lipids as well as a decreased peak of myoinositol are reliable data for the diagnosis of glioblastoma [30,32] and are helpful to discriminate glioblastomas from metastasis, lymphoma and brain abscess. As suggested by some authors, these explorations are also useful to assess the peritumoral degree of invasion and may be used as a guidance for biopsy [33,34] and to monitor disease evolution after treatment.

#### 3.2.4. Basic and Molecular Pathology

The diagnosis of glioblastoma is easily made on surgical resections or biopsy samples. Glioblastomas are high grade gliomas, grade IV according to the World Health Organization (WHO) classification of the central nervous system tumors [35,36]. Glioblastoma is composed of poorly differentiated, often pleomorphic tumor cells with predominant astrocytic differentiation [37]. Histopathological features include nuclear atypia, cellular pleomorphism, high mitotic activity, vascular thrombosis, microvascular proliferation and necrosis [28]. Since the 2016 WHO classification of the central nervous system tumors [36] and advances in immunohistochemistry, glioblastomas are now defined by their Isocitrate dehydrogenase (*IDH*) status dividing this entity into glioblastoma *IDH*-mutant or *IDH*-wild type. The latter is, by far, the most frequent accounting for 90% of cases and is predominant in patients over 55 years of age. Glioblastomas *IDH*-mutant (10% of cases) is predominant in younger patients, often results from the transformation of a lower-grade glioma and is associated with longer median survival [38]. Complementary immunohistochemistry and molecular techniques are now routinely used for diagnostics and prognostic purposes. (*IDH*) alterations as well as hypermethylation of the O^6^-methylguanine-DNA methyltransferase (*MGMT*) gene promoter [39] are predictive of longer survival contrary to Telomerase Reverse Transcriptase promoter (*TERT*p) variants and chromosome 10 deletion which are poorer prognostic factors. Gain of function variants in the *P53* gene [40] and epidermal growth factor receptor (*EGFR*) gene alterations [41] still are of uncertain prognostic significance. *MGMT* is involved in DNA repair and its expression is associated with drug resistance including temozolamide, the most frequent first-line chemotherapy used in a context of glioblastoma. As a consequence, hypermethylation of the *MGMT* gene promoter is a valuable predictive marker and is associated with longer overall survival and longer progression-free survival [42]. More recently, it has been proven [43] that variants in the tumor suppressor gene Phosphatase and tensin homolog (*PTEN*) or the loss of chromosome 10 is involved in the glioblastoma oncogenesis.

### 3.3. Current Management

#### 3.3.1. Surgical Procedure

Whenever possible, the first step consists in complete macroscopic surgical resection. Literature data suggest that a resection > 90% of the contrast enhancement of the lesion in patients with no comorbidities improves the patient outcome at the time of diagnosis and recurrence [44,45,46]. Surgical resection is usually proposed to patients under the age of 70 in good condition (Karnofsky scale > 70) [46] and a tumor accessible to complete removal. Otherwise, surgical debulking or stereotactic biopsy is performed to confirm the diagnosis before adjuvant therapy [47,48]. Due to the importance of a complete resection on survival, advances in surgical techniques have been made [49,50] such as awake craniotomy or neuromonitoring [51] to improve the resection quality and prevent subsequent deficits. Furthermore, fluorescence-guided surgery has been developed to guide the resection with better outcomes on resection and progression-free survival [52,53,54]. More recently, resection devices have evolved with the use of Laser-Interstitial Thermal Therapy (LITT) [55], which provides a less invasive, percutaneous approach through the insertion of an optical fiber. The generated thermal injury induces tumor necrosis [55,56,57]. Another novelty in the surgical resection of brain tumors is mass spectrometry-based intraoperative monitoring of tumor metabolites. Cell content is analyzed allowing for an accurate and molecular delineation of tumor margins and thus for an optimal tumor resection [58,59]. A summary of clinical presentations, radiology, biology and treatments together with their interactions is illustrated in Figure 3.

#### 3.3.2. Medical Treatment of Glioblastomas

The standard of care for patients aged less than 70 relies on radiotherapy (RT) and adjuvant temozolomide. This protocol improved the overall survival in a large randomized phase III trial [60]. Radiotherapy is given for a six-week period with a total dose of 60 grays. Temozolomide is an alkylating agent administered daily during the RT and then, for six cycles of five consecutive days per month, one month after the end of the RT. The absence of hypermethylated *MGMT* promoter [61] is a negative prognostic and predictive factor of temozolomide efficiency. The treatment protocol proposed by Stupp allows for increasing the average survival rate from 12.1 months using RT alone to 14.6 months, and the two-year survival rate from 8 to 26% with concomitant temozolomide [60]. This randomized controlled trial did not include patients older than 70. In the latter population, the standard of care is based on hypofractionated radiotherapy and temozolomide [62] whenever feasible, but the treatment depends on the patient’s general condition. RT alone (54 grays) has been proposed with a positive impact on survival (29.1 weeks compared to 16.9 weeks in patients with supportive care alone) and with no alteration of the quality of life [63]. These results have been validated in patients with a Karnofsky [64] performance status (KPS) > 60. More recently, some authors have shown that short-course RT plus temozolomide was associated with longer survival (9.3 versus 7.6 months) in older patients (>65 years). Malmstrom et al. [65] randomized patients aged 60 years and older presenting with a glioblastoma to assess the optimal palliative treatment. The conclusion of the trial was that radiotherapy alone is associated with poor outcomes. On the contrary, both temozolamide and hypofractionated radiotherapy appeared as standards of care especially in patients with methylation of the MGMT gene promoter. In patients with poor general health, supportive care may be proposed to preserve the quality of life with the shortest length of hospital stay if possible [62].

The disease progression is evaluated using brain MRI every 2 to 3 months according to the response assessment in Neuro-oncology (RANO) criteria [66]. At the time of recurrence, there is no standard of care. The main determinants for treatment proposals are the patients’ general condition and treatments previously administered [67]. A second surgery can be proposed in young patients with preserved KPS. This strategy has been shown to be associated with longer survival in selected patients [68] (14 months versus 22 months of overall survival in patients with second surgery at recurrence). Eighty percent of patients did not need rehabilitation after a second surgery. In selected patients, the use of intracavitary carmustine wafers (BCNU) has been proposed at initial surgery or in case of recurrence [69]. Efficiency and adverse effects of this therapy remains a matter of debate [70,71]. In the absence of safe possible resection, a second line chemotherapy can be proposed including nitrosoureas, temozolomide or antiangiogenic drugs such as bevacizumab but with no clear results in terms of benefit to date [72] and poor outcomes with an average overall survival rate of 6 months from recurrence [73]. Due to the poor prognosis of glioblastomas, there is an urgent need for new therapies. The REGOMA trial (Regorafenib in Relapsed Glioblastoma) [74] is a phase II randomized trial. The aim was to propose the use regorafenib, an inhibitor of angiogenic and oncogenic receptor of tyrosine kinases at the recurrence of glioblastomas. The results were rather positive on overall survival and with few side effects. A phase III trial is expected.

### 3.4. Current Research and Perspectives

#### 3.4.1. Omics Approaches

Over the last decade, precision medicine, also through omics approaches (Figure 4), has offered new insights in the diagnosis and management of glioblastomas [75,76]. In the past few years, radiomics brought significant insights in the characterization and predictive models of glioblastomas [77,78]. Radiomics is based on the extraction of a large amount of data from medical images. Radiomics is then enriched with clinical, genomics and proteomics data to establish new diagnosis and prognosis criteria to enhance treatment efficiency. Other promising areas in the fight against glioblastoma are the genomics and transcriptomics approaches. The emergence of big data in the precision medicine offered new therapeutic perspectives. Through ambitious projects such as the human genome project [79], genomics helped us in improving the understanding of glioblastoma. In the continuation of the project, RNA sequencing-based genes proposed genomics signature of life expectancy in patients with glioblastomas [80]. Transcriptomics and other omics technics proposed prognostic tools for the comprehension of the disease [81,82]. Along with these techniques, liquid chromatography and mass spectrometry analyses from different samples such as CSF (Cerebrospinal fluid), urine, blood or glioblastoma cell lines [83] have provided a comprehensive view of the altered metabolic pathways in patients with glioblastoma [84,85]. More recently, special attention has been given to alterations of lipid metabolism in glioblastomas. Based on omics human studies, Guo et al. have described a decrease of 90% of lipid levels in tumor tissue except for phosphatidylcholine and cholesterol ester levels which appeared high in glioma tissue while they are absent in normal brain tissue [86]. These data may emphasize the key role of certain lipids in the glioblastoma metabolism to facilitate tumor growth. Moreover, the metabolic signature of brain tumors in the plasma is of interest for the grading and prognosis of these diseases. It has been shown that the plasma level of metabolites of interest can help to define the grading of brain glioma and to provide prognostic information in patients with a similar glioma grade [87]. Nevertheless, data are still rare for these emerging approaches; however, the promise of precision medicine and the surge of multimodal data-driven strategies can provide valuable tools for the development of biomarkers and innovative therapies in glioblastomas [88].

#### 3.4.2. Novel Therapies for Glioblastomas

Due to the adverse outcome in patients with glioblastoma and the high frequency of this disease, innovative therapies are being tested in different randomized controlled trials [73,89,90]. With the development of a better understanding of molecular pathways triggering glioblastoma growth [27,28,29], the traditional approach of antitumor therapy is being progressively complemented by a more personalized approach [91,92]. Treatment schedules have been rethought as well as the drugs themselves. Two major drawbacks consist in the difficulty for most of the drugs is to pass through the blood–brain barrier (BBB) and to target tumor cells due to the presence of abnormal vessels and necrosis, which hampers drugs being delivered at a suitable concentration. Some emerging techniques have been proposed to improve the distribution of antitumor therapy, notably the conjugation of drugs with protein to facilitate the movement across the BBB and specifically target the tumor [93], the use of convection-enhanced delivery consisting in the direct administration inside the tumor via a catheter [94] and the use of nanoparticles. The increase in BBB permeability during chemotherapy administration via focused ultrasounds is being also tested [95]. Meanwhile, immunotherapy approaches are known for a long time, with promising results in many cancers such as melanoma but deceiving results in patients with gliomas [96]. Cancer vaccination has recently been proposed referring to the activation of an immune response against tumor antigens. These new technologies have been applied for glioblastoma treatment with different vectors [97,98,99]. Two modalities have been tested: peptide vaccines targeting EGFR, IDH1 or heat shock proteins, and cell-based vaccines consisting in the injection of ex vivo modified cells, mainly dendritic cells [100]. Despite encouraging results in animal models in terms of disease control [101], cancer vaccination in glioblastomas has not yet proven its efficacy on overall survival in phase III studies [9]. Recently, the role of tumor-associated macrophages (TAM) has been highlighted in the genesis and resistance to treatment of glioblastoma cells [102,103,104]. Landry et al. [102] showed that the TAM located in the core have different characteristics and metabolic pathways compared to those located in the periphery of the glioblastoma. For these reasons, they reaffirm the need for a multi-targeted approach through a modulation of the TAM. Furthermore, tumor-associated neutrophils (TAN) are found to be involved in necrosis onset in glioblastoma patients [105,106]. In this context, the mechanism of necrosis could be a neutrophil-mediated ferroptosis. The latter could have a pro-tumorigenic role [105]. Thus, targeted therapies are potential novel therapies for glioblastomas to prevent TAN recruitment.

Stupp et al. recently developed a new therapeutic modality in the treatment of recurrent glioblastomas consisting in the local delivery of low-intensity electric fields via a non-invasive transducer [73]. The device (NovoTTF-100A) was tested in a phase III study and randomized with active chemotherapy as an alternative arm. The overall survival was similar in both arms (6 months) with fewer adverse events in the NovoTTF-100A group and a better quality of life. For this reason, the tumor-treating field is considered as a standard of care in some guidelines [107]. Moreover, in the last years, the association of molecularly targeted drugs such as tyrosine kinase administration or others combined with X-rays to decrease radioresistance due to hypoxia showed encouraging results [108,109,110].

**Figure 4 jpm-11-00258-f004:**
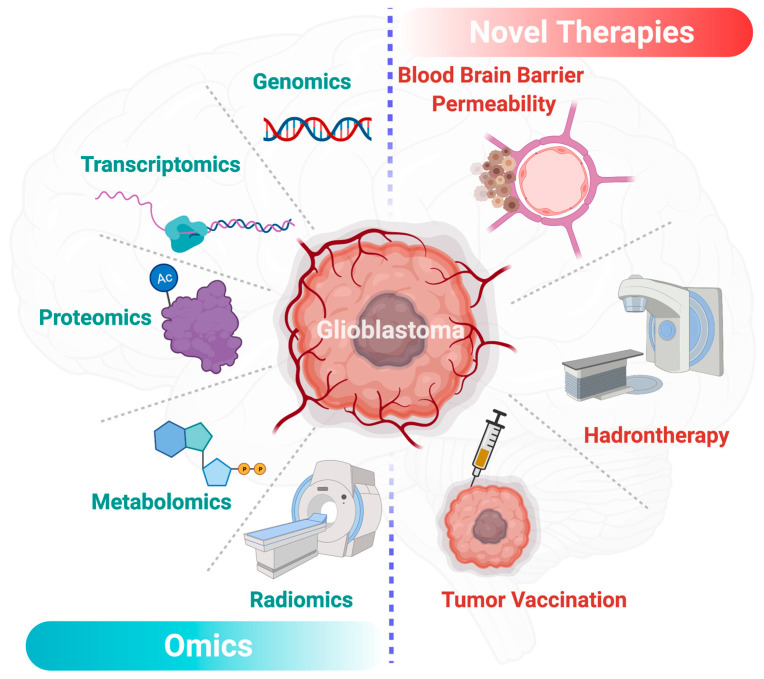
Overview of the main driving omics technologies and therapeutic perspectives for glioblastoma in the precision medicine era. This figure has been created with BioRender.com (accessed on 9 March 2021).

## 4. Conclusions

The authors are aware of the limits of using such automatic topic literature search tools. This limits might be related to recency or coverage scope of such tools that might lead to some key literature outputs being missing. This highlights the importance of manual curation combined with the use of multiple tools to consistently cover the area of interest.

In conclusion, despite a better understanding of the molecular pathways leading to glioblastoma development and growth, outcomes remain poor in terms of survival. Many trials are currently in progress with new therapeutic approaches to go from a global approach to a more personalized approach. The main objective of these approaches is to deliver the chosen drug inside the tumor and to adapt its concentration to the tumor characteristics. Then, the modality of drug delivery has become as challenging as the drug itself.

## Figures and Tables

**Figure 1 jpm-11-00258-f001:**
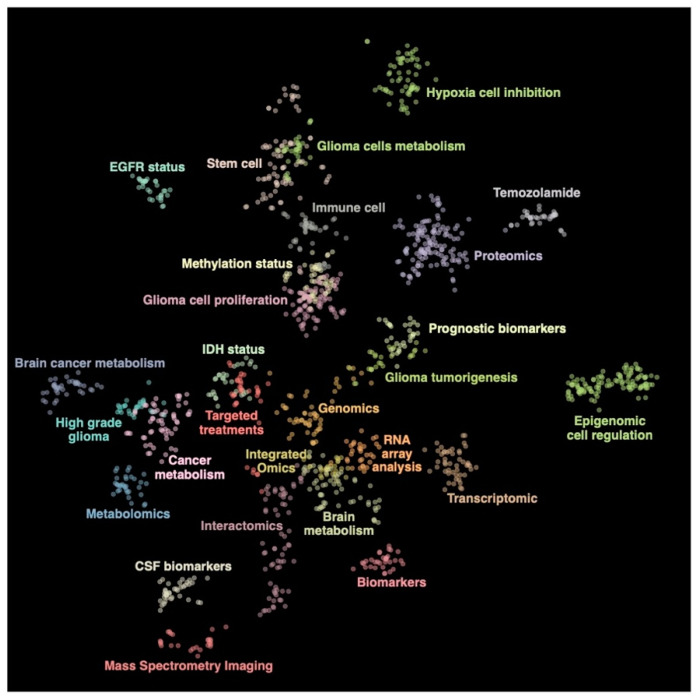
Topic representation of the included literature related to glioblastoma. The figure highlights twenty-seven clusters.

**Figure 2 jpm-11-00258-f002:**
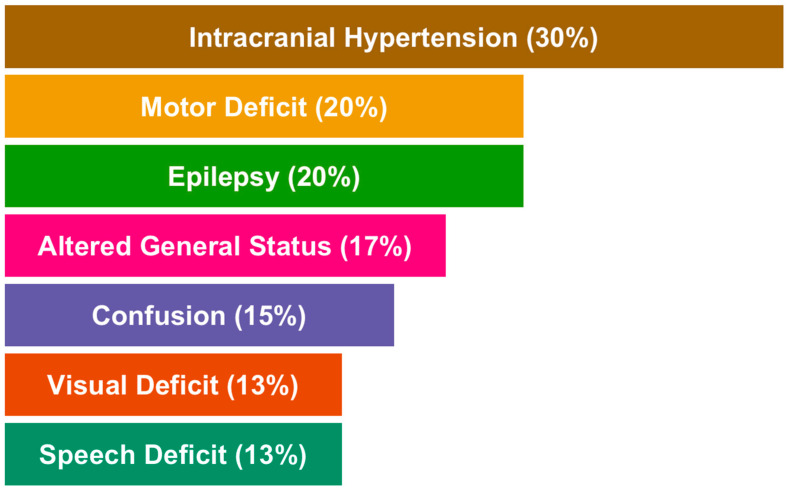
Overview of the main reported clinical features in glioblastomas.

**Figure 3 jpm-11-00258-f003:**
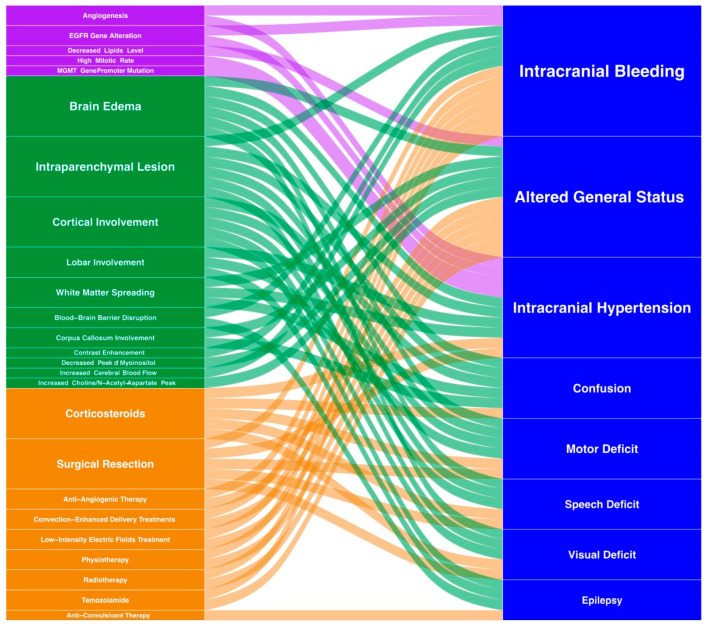
Integrative visualization summary of main clinical symptoms and signs and their interactions with radiology, biology and treatment features. The box and annotation sizes are proportional to the item frequency. Clinical: blue, Biology: purple, Radiology: green, Treatment: orange.

## Data Availability

Data supporting the finding are presented in the text and Appendix A.

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
