# Peer review of "Diagnosis and Management of Glioblastoma: A Comprehensive Perspective"

_jpm, 2021, doi:10.3390/jpm11040258_

Round 1

Reviewer 1 Report

The authors have fully addressed all concerns suggested to improve the manuscript. In my opinion, the revised manuscript is now suitable for publication in present form.

Author Response

Rouen, March 23th, 2021

Dear Editor,

We would like to acknowledge the reviewers for their comments and the editor for giving us the opportunity to revise our manuscript referenced as jpm-1157407 by Gilard et al.

The positive and constructive comments of the reviewers have been considered and we think that our manuscript has been markedly improved.

Please find below the detailed answers to the reviewers’ concerns.

The main changes in the text are highlighted in red in the manuscript.

We hope that our manuscript is now acceptable for publication in the Journal of Personalized Medicine.

Sincerely yours,

Dr. Vianney GILARD, MD

Department of Neurosurgery

Rouen University Hospital

7 Boulevard Gambetta, 76000 Rouen, France.

E-mail: vianney.gilard@chu-rouen.fr

Reviewer #1

The authors have fully addressed all concerns suggested to improve the manuscript. In my opinion, the revised manuscript is now suitable for publication in present form.

We thank the reviewer for this kind comment.

Reviewer 2 Report

very nice review. 

Author Response

Rouen, March 23th, 2021

Dear Editor,

We would like to acknowledge the reviewers for their comments and the editor for giving us the opportunity to revise our manuscript referenced as jpm-1157407 by Gilard et al.

The positive and constructive comments of the reviewers have been considered and we think that our manuscript has been markedly improved.

Please find below the detailed answers to the reviewers’ concerns.

The main changes in the text are highlighted in red in the manuscript.

We hope that our manuscript is now acceptable for publication in the Journal of Personalized Medicine.

Sincerely yours,

Dr. Vianney GILARD, MD

Department of Neurosurgery

Rouen University Hospital

7 Boulevard Gambetta, 76000 Rouen, France.

E-mail: vianney.gilard@chu-rouen.fr

Reviewer #2       

very nice review. 

 We thank the reviewer for this kind remark.

Reviewer 3 Report

In this manuscript, the authors have presented a current status of diagnosis and management of glioblastoma. The article could be further improved by addressing the following points:

  1. Immune cells, such as macrophages and neutrophils, have recently been shown to play important roles in glioblastoma progression and could be targeted for therapeutics. This manuscript should add a part to discuss this aspect.
  2. Page 6, Figure 3 looks odd because a large white area in the middle.
  3. Some typo errors need to be fixed. e.g. page 8, Line 298, “based-cell”.

Author Response

Rouen, March 23th, 2021

Dear Editor,

We would like to acknowledge the reviewers for their comments and the editor for giving us the opportunity to revise our manuscript referenced as jpm-1157407 by Gilard et al.

The positive and constructive comments of the reviewers have been considered and we think that our manuscript has been markedly improved.

Please find below the detailed answers to the reviewers’ concerns.

The main changes in the text are highlighted in red in the manuscript.

We hope that our manuscript is now acceptable for publication in the Journal of Personalized Medicine.

Sincerely yours,

Dr. Vianney GILARD, MD

Department of Neurosurgery

Rouen University Hospital

7 Boulevard Gambetta, 76000 Rouen, France.

E-mail: vianney.gilard@chu-rouen.fr

Reviewer 3:       

In this manuscript, the authors have presented a current status of diagnosis and management of glioblastoma. The article could be further improved by addressing the following points:

  1. Immune cells, such as macrophages and neutrophils, have recently been shown to play important roles in glioblastoma progression and could be targeted for therapeutics. This manuscript should add a part to discuss this aspect.

These elements have been added in the paragraph on “novel therapies for glioblastomas” and updated as suggested. Page 9, lines 292-300.

  1. Page 6, Figure 3 looks odd because a large white area in the middle.

The Figure 3 has been edited accordingly.

  1. Some typo errors need to be fixed. e.g. page 8, Line 298, “based-cell”.

The modification has been made accordingly.

Round 2

Reviewer 3 Report

The comments have been addressed. 

This manuscript is a resubmission of an earlier submission. The following is a list of the peer review reports and author responses from that submission.

Round 1

Reviewer 1 Report

This is a very well written paper which gives a good overview on current treatment strategies of glioblastomas.

I have nothing to add. 

Author Response

Reviewer #1

This is a very well written paper which gives a good overview on current treatment strategies of glioblastomas. I have nothing to add. 

We thank the reviewer for this kind comment.

Reviewer 2 Report

I read with interst the work by Gilard et colleagues, presenting a review on diagnosis and management of glioblastoma performed thanks to an interesting semiautomated literature analysis. This work could be suitable for publication.

Specific issues to be addressed:

- in the abstract, replace “14” with “15” months, since it is a better approximation of the traditional 14.6 months figure in the work by Stupp.

- Radiological characteristics: necrosis is not necessarily hypointense in T2-weighted images. Probably the authors meant: T1.

- In the same paragraph, spectroscopy data are not so “specific”, since inversion of Cho/cre and Cho/NAA ratios are linked to increased cellular turnover, whereas increased lipid/lactate peaks is linked to necrosis.

- again in radiological paragraph, assessment of role of diffusion-weighted images is missing.

- Reference 32: the “real” blue book should be cited instead (or in addition to) the commenting article in Acta Neuropathologica.

- Basic and molecular pathology: this paragraph needs profound revision. IDH status, as assessed by IHC and/or gene sequencing, is fundamental in distinguishing glioblastoma, IDH-wildtype from glioblastoma, IDH-mutated, which are distinct entities according to WHO 2016. Methylation of MGMT promoter has a fundamental prognostic and predictive role. Other assessments are complementary (e.g., p53, TERTp, EGFR, or also PTEN, CDKN2A/B, or ch10 deletion). ATRX mutation and 1p/19q codeletion have a role in lower-grade glioma, not in glioblastoma.

- Medical treatment: Pag. 6, line 179, concomitant AND ADJUVANT temozolomide. Line 183, “[…] is a NEGATIVE prognostic AND PREDICTIVE factor of temozolomide efficiency”. The subsequent sentence, i.e. “RT alone may be therefore proposed” should be removed because it is not true, except in the contest of randomized trials.

- Medical treatment in elderly: according with NORDIC trial, temozolomide alone in patients with methylated MGMT promoter can be proposed.

- Among drugs used for recurrence, regorafenib must be cited, based on results of the REGOMA trial.

- Omics approach: first line, “precision medicine, ALSO through omics approach”. Indeed, many different approaches have been used in precision medicine.

- Same paragraph, among materials analyzed, glioblastoma cell lines could be cited (see e.g. PMID 26857460)

- Novel therapies for glioblastoma: TTF has been validated also for glioblastoma at first diagnosis (PMID 29260225) and it is considered as a standard in many guidelines.

- A “limitation” paragraph must be added: in my opinion, though interesting, the automated way of analyzing literature that has been employed limits the ability of the Authors to identify some key issues or promising fields in glioblastoma research.

- minor English inaccuracies should be corrected.

Author Response

Rouen, February 26th, 2021

Dear Editor,

We would like to acknowledge the reviewers for their comments and the editor for giving us the opportunity to revise our manuscript referenced as jpm-1125994 by Gilard et al.

The positive and constructive comments of the reviewers have been considered and we think that our manuscript has been markedly improved.

Please find below the detailed answers to the reviewers’ concerns.

The main changes in the text are highlighted in red in the manuscript.

We hope that our manuscript is now acceptable for publication in the Journal of Personalized Medicine.

Sincerely yours,

Dr. Vianney GILARD, MD

Department of Neurosurgery

Rouen University Hospital

7 Boulevard Gambetta, 76000 Rouen, France.

E-mail: vianney.gilard@chu-rouen.fr

Reviewer #2       

I read with interest the work by Gilard et colleagues, presenting a review on diagnosis and management of glioblastoma performed thanks to an interesting semiautomated literature analysis. This work could be suitable for publication.

Specific issues to be addressed:

- in the abstract, replace “14” with “15” months, since it is a better approximation of the traditional 14.6 months figure in the work by Stupp.

We thank the reviewer for this kind remark.

This has been corrected. Page 1, line 28.

- Radiological characteristics: necrosis is not necessarily hypointense in T2-weighted images. Probably the authors meant: T1.

This has been corrected. Page 4, line 135.

- In the same paragraph, spectroscopy data are not so “specific”, since inversion of Cho/cre and Cho/NAA ratios are linked to increased cellular turnover, whereas increased lipid/lactate peaks is linked to necrosis.

We thank the reviewer for highlighting this.

The text has been edited accordingly. Page 4, lines 143-145.

- Again in radiological paragraph, assessment of role of diffusion-weighted images is missing.

Indeed, diffusion-weighted images can allow useful information concerning the grading of brain tumors. This point has been added. Page 4, lines 137-139.

- Reference 32: the “real” blue book should be cited instead (or in addition to) the commenting article in Acta Neuropathologica.

The blue book citation has been added as reference number 36. Page 5, line 154.

- Basic and molecular pathology: this paragraph needs profound revision. IDH status, as assessed by IHC and/or gene sequencing, is fundamental in distinguishing glioblastoma, IDH-wildtype from glioblastoma, IDH-mutated, which are distinct entities according to WHO 2016. Methylation of MGMT promoter has a fundamental prognostic and predictive role. Other assessments are complementary (e.g., p53, TERTp, EGFR, or also PTEN, CDKN2A/B, or ch10 deletion). ATRX mutation and 1p/19q codeletion have a role in lower-grade glioma, not in glioblastoma.

The paragraph has been modified accordingly on page 5, lines 157-175.

- Medical treatment: Pag. 6, line 179, concomitant AND ADJUVANT temozolomide. Line 183, “[…] is a NEGATIVE prognostic AND PREDICTIVE factor of temozolomide efficiency”. The subsequent sentence, i.e. “RT alone may be therefore proposed” should be removed because it is not true, except in the contest of randomized trials.

This has been corrected accordingly on page 6, line 204.

- Medical treatment in elderly: according with NORDIC trial, temozolomide alone in patients with methylated MGMT promoter can be proposed.

As suggested, we have discussed the opportunity to treat elderly patients with glioblastoma accordingly. Page 7, lines 219-223.

- Among drugs used for recurrence, regorafenib must be cited, based on results of the REGOMA trial.

As suggested, we have discussed the results of the REGOMA trial.

Page 7, lines 238-242.

- Omics approach: first line, “precision medicine, ALSO through omics approach”. Indeed, many different approaches have been used in precision medicine.

This sentence has been edited accordingly. Page 7, line 246.

- Same paragraph, among materials analyzed, glioblastoma cell lines could be cited (see e.g. PMID 26857460)

This has been added as suggested. Page 8, line 259.

- Novel therapies for glioblastoma: TTF has been validated also for glioblastoma at first diagnosis (PMID 29260225) and it is considered as a standard in many guidelines.

This has been added. Page 8, lines 299-300.

- A “limitation” paragraph must be added: in my opinion, though interesting, the automated way of analyzing literature that has been employed limits the ability of the Authors to identify some key issues or promising fields in glioblastoma research.

This has been added. Page 9, lines 307-310.

- minor English inaccuracies should be corrected.

The text has been checked and edited accordingly.

Reviewer 3 Report

The review paper entitled: “Diagnosis and Management of Glioblastoma: a Comprehensive Perspective”, deals with a topic of strong clinical impact considering the high aggressiveness of Glioblastoma and the difficulty of finding the best cure treatment to date that is effective by increasing life expectancy and reducing the mortality rate.

However, although the topic is interesting, the authors should implement the manuscript to improve  and make it suitable for publication.

The main changes that need to be addressed by the authors are the following:

The introduction is very poor and needs to be improved summarizing the main clinical and therapeutic issues of Glioblastoma, also in support of the need to get a more complete overview of Glioblastoma through this review.

As regards the section “Current researches and perspectives”, in particular the paragraph “Omics approaches” is very poor and should be improved. , The radiomics is missing and needs to be added. See for example: https://pubmed.ncbi.nlm.nih.gov/31165039/ .

The genomic and transcriptomic approaches are missing  and should be added and described. See for example: https://pubmed.ncbi.nlm.nih.gov/30796273/  https://pubmed.ncbi.nlm.nih.gov/32453707/  https://pubmed.ncbi.nlm.nih.gov/33126510/  

As regards the paragraph “Novel therapies for glioblastomas”, an overview of new therapeutic strategies, such as hadrontherapy and combined treatments with targeted drugs and molecules are missing and should be added and described. See for example: https://pubmed.ncbi.nlm.nih.gov/33020459/; https://pubmed.ncbi.nlm.nih.gov/31554327/  https://pubmed.ncbi.nlm.nih.gov/32486205/ .

A new representative figure might be useful to complete the sections “Current researches and perspectives" and “Novel therapies for glioblastomas”.

The reference list should be implemented with those suggested or even with other bibliographies to support the sections to be improved.

Author Response

Rouen, February 26th, 2021

Dear Editor,

We would like to acknowledge the reviewers for their comments and the editor for giving us the opportunity to revise our manuscript referenced as jpm-1125994 by Gilard et al.

The positive and constructive comments of the reviewers have been considered and we think that our manuscript has been markedly improved.

Please find below the detailed answers to the reviewers’ concerns.

The main changes in the text are highlighted in red in the manuscript.

We hope that our manuscript is now acceptable for publication in the Journal of Personalized Medicine.

Sincerely yours,

Dr. Vianney GILARD, MD

Department of Neurosurgery

Rouen University Hospital

7 Boulevard Gambetta, 76000 Rouen, France.

E-mail: vianney.gilard@chu-rouen.fr

Reviewer 3:       

The review paper entitled: “Diagnosis and Management of Glioblastoma: a Comprehensive Perspective”, deals with a topic of strong clinical impact considering the high aggressiveness of Glioblastoma and the difficulty of finding the best cure treatment to date that is effective by increasing life expectancy and reducing the mortality rate.

However, although the topic is interesting, the authors should implement the manuscript to improve and make it suitable for publication.

The main changes that need to be addressed by the authors are the following:

The introduction is very poor and needs to be improved summarizing the main clinical and therapeutic issues of Glioblastoma, also in support of the need to get a more complete overview of Glioblastoma through this review.

The introduction has been edited accordingly. We highlighted the heterogeneity of molecular characteristics, the poor outcomes of the existing treatments of glioblastoma and the need for need approach with its preliminary results.

Page 2, lines 48-56.

As regards the section “Current researches and perspectives”, in particular the paragraph “Omics approaches” is very poor and should be improved. The radiomics is missing and needs to be added. See for example: https://pubmed.ncbi.nlm.nih.gov/31165039/ .

The section has been edited accordingly. Pages 7-8, lines 247-252.

The genomic and transcriptomic approaches are missing and should be added and described.

The role of genomic and transcriptomic approaches was added and discussed as requested. Page 8, lines 251-257.

As regards the paragraph “Novel therapies for glioblastomas”, an overview of new therapeutic strategies, such as hadrontherapy and combined treatments with targeted drugs and molecules are missing and should be added and described. See for example: https://pubmed.ncbi.nlm.nih.gov/33020459/; https://pubmed.ncbi.nlm.nih.gov/31554327/  https://pubmed.ncbi.nlm.nih.gov/32486205/ .

The paragraph on “novel therapies for glioblastomas” has been updated as suggested. Page 9, lines 300-302.

A new representative figure might be useful to complete the sections “Current researches and perspectives" and “Novel therapies for glioblastomas”.

Figure 4 has been added describing current research and therapeutic axes.

The reference list should be implemented with those suggested or even with other bibliographies to support the sections to be improved.

The suggested references have been added.

Round 2

Reviewer 3 Report

The authors have fully addressed all concerns suggested to improve the manuscript.